# Critical Review of Nanoindentation-Based Numerical Methods for Evaluating Elastoplastic Material Properties

**Xu Long** [1], **Ruipeng Dong** [1], **Yutai Su** [1,*] and **Chao Chang** [2,*]

1   School of Mechanics, Civil Engineering and Architecture, Northwestern Polytechnical University, Xi'an 710072, China; xulong@nwpu.edu.cn (X.L.)
2   School of Applied Science, Taiyuan University of Science and Technology, Taiyuan 030024, China
*   Correspondence: suyutai@nwpu.edu.cn (Y.S.); cc@tyust.edu.cn (C.C.)

**Abstract:** It is well known that the elastoplastic properties of materials are important indicators to characterize their mechanical behaviors and are of guiding significance in the field of materials science and engineering. In recent years, the rapidly developing nanoindentation technique has been widely used to evaluate various intrinsic information regarding the elastoplastic properties and hardness of various materials such as metals, ceramics, and composites due to its high resolution, versatility, and applicability. However, the nanoindentation process of indenting materials on the nanoscale provides the measurement results, such as load-displacement curves and contact stiffness, which is challenging to analyze and interpret, especially if contained in a large amount of data. Many numerical methods, such as dimensionless analysis, machine learning, and the finite element model, have been recently proposed with the indentation techniques to further reveal the mechanical behavior of materials during nanoindentation and provide important information for material design, property optimization, and engineering applications. In addition, with the continuous development of science and technology, automation and high-throughput processing of nanoindentation experiments have become a future trend, further improving testing efficiency and data accuracy. This paper critically reviewed various numerical methods for evaluating elastoplastic constitutive properties of materials based on nanoindentation technology, which aims to provide a comprehensive understanding of the application and development trend of the nanoindentation technique and to provide guidance and reference for further research and applications.

**Keywords:** indentation; elastoplastic; load-displacement curves; contact stiffness; dimensionless; machine learning; finite element model

## 1. Introduction

Elastoplastic and plasticity measurement of materials is an essential research area in the mechanics of materials. In modern engineering techniques, it is vital to accurately measure the elastic and plastic properties of materials. The early experiments mainly used tensile, impact, and hardness tests to study the physical properties and mechanical properties of the materials under load [1–6]. However, these methods cannot directly measure the nanoscale and mesoscale mechanical properties of materials [7]. With the rapid development of devices for load and displacement monitoring, nanoindentation techniques have emerged, which are more advanced than traditional tensile or compression experiments and are widely used to perform the elastoplastic measurements of the constitutive behavior of materials [8–13].

The invention of nanoindentation can be traced back to the mid-1950s when scholars in the Union of Soviet Socialist Republics started to record the load-depth curves of different metals and minerals applied to nanoscale depth sensing technology. Subsequently, this technology developed rapidly and was widely used around the world. Particularly, it should be noted that continuous explorations have been made to achieve further development of methods to interpret test data and apply them to the estimation of mechanical

properties of materials [14]. In 1992, Venkataraman et al. [15] first applied nanoindentation to measure materials in a given set of systems. In 1966, Bolshakov et al. [16] presented the first theoretical model regarding nanoindentation, proposing an indentation theory based on contact mechanics and elastoplasticity mechanics for describing the mechanical properties of solid materials, such as hardness and elastic modulus. In addition, many other scientists and engineers have made outstanding contributions and great progress in developing nanoindentation technology.

Nowadays, nanoindentation has become one of the effective methods for material elastoplasticity measurements and therefore is widely adopted in various fields, such as thin-film biomaterials, semiconductor materials, metals, polymers, and ceramics [17–19]. In today's science and technology field, the wide application of the above materials is undeniable. However, the unique application value of thin films is remarkable and has led to in-depth research by many scholars. Not only has the investigation of film formation been widely explored [20,21], but the measurement of the mechanical properties of thin films has also become an integral part of the development of thin-film technology. The advantage of atomic force microscopy (AFM)-nanoindentation has also greatly simplified the nanoindentation experimental procedure and improved the experimental efficiency [18,19,22,23]. With the development of nanoscience and technology, the nanoindentation technique will certainly be further improved in terms of measurement accuracy and be more extensively applied in future engineering applications. Exploration of elastoplastic material parameters by indentation has become a major focus in recent works on indentation. Nevertheless, the fundamental question of whether the elastoplastic properties of a material sample in the elastic and strain hardening stages, as shown in Figure 1, can be uniquely determined remains challenging.

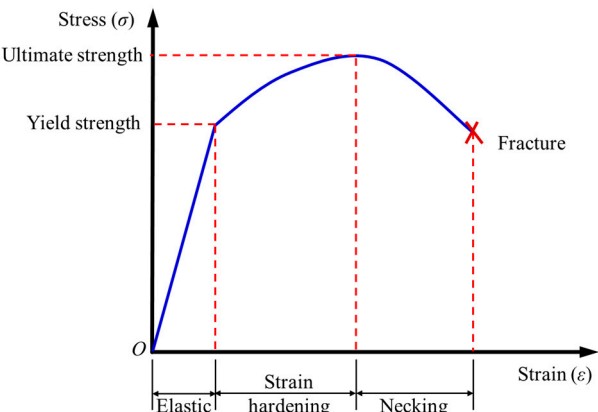

**Figure 1.** Different stages of the stress-strain curve of materials, including the elastic stage, strain hardening stage, and necking stage [24].

This paper focuses on the material-related properties obtained through nanoindentation, specifically the stress-strain relationship of the material. The recent studies on nanoindentation measurement of elastoplasticity are critically reviewed, focusing on the load-displacement (*P–h*) curve measured by nanoindentation and also the contact stiffness combined with dimensionless, machine learning, density functional theory, molecular dynamics, and finite element model update (FEMU) methods. To solve the material elastoplastic properties, the flowchart of the solution steps is demonstrated in Figure 2. Finally, the application, development, and prospect of nanoindentation in practical engineering are further discussed.

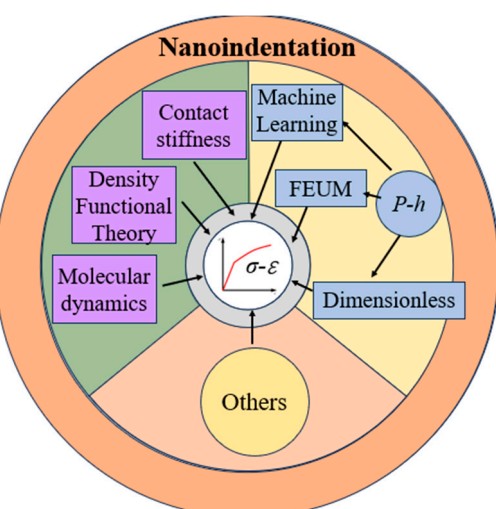

**Figure 2.** Schematic diagram of nanoindentation method for measuring mechanical properties of materials.

## 2. Load-Displacement Curve

The study of the nanoindentation of materials to obtain load-displacement curves has been published in the previous results. In the early 1970s, experimental tools for load and displacement measurements were started, and the indentation load-displacement curve shown in Figure 3 was obtained using an instrumented microhardness tester [25]. With the continuous development of instrumentation and nanoindentation theories, instrumented nanoindentation was used to obtain the *P–h* curves of materials, which built a foundation for realizing various property measurements and applications at the nanoscale [15,25]. With the continued improvement and development of nanoindentation, there are different indenter shapes used for experiments. The shape of the indenter has an important impact on nanoindentation experiments, and different indenter shapes can provide different test information and applications. In fact, the selection of the appropriate indenter shape depends on the material properties and testing needs. In addition to using Berkovich indenters for indentation, various other forms are employed. For instance, cube corner, Vickers, and Knoop nanoindenters are widely recognized standards due to their broad applicability and recognized characteristics. Furthermore, researchers are currently exploring similar geometries to expand the range of options available [26,27]. Later, Giannakopoulos and Suresh obtained *P–h* curves based on finite element (FE) simulations to simulate the experimental process of nanoindentation, which provided a more intuitive understanding of the nature of materials at the macroscopic and microscopic levels and made an important contribution to the accurate prediction of material elastoplastic [16]. The results by Li et al. [28] showed that the *P–h* curve could be generated from a single indentation depth when performing indentation tests on poly (methyl methacrylate) (PMMA) films using a Berkovich indenter, regardless of the depth of the indentation. In addition, they performed finite element simulations by using the Oyen–Cook intrinsic model and succeeded in reproducing the *P–h* curves obtained from experimental measurements with reasonable accuracy. The *P–h* curves due to the nanoindentation of various materials have been continuously investigated, and the elastoplastic properties have been further explored by different numerical and theoretical methods, such as reverse algorithms, machine learning, etc.

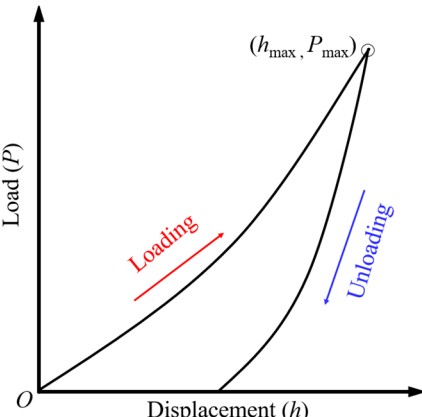

**Figure 3.** Load-displacement curve where $h_{max}$ represents the maximum indentation depth and $P_{max}$ represents the maximum load corresponding to the maximum indentation depth [29].

### 2.1. Dimensionless Method

In recent years, the dimensionless method has attracted extensive academic attention in the field of nanoindentation, and many researchers have conducted several cutting-edge works to reveal the relationship between the mechanical properties of materials and geometric scales through exploring the dimensionless process of nanoindentation experimental data [30,31]. Their researches provide an important foundation and theoretical guidance for an in-depth understanding of the mechanical behavior at the nanoscale and its application in material designs and characterizations [32–34]. The research results in this field offer a broader prospect for future research on the mechanics of materials and the development of nanotechnology. Dao et al. [35] used dimensional analysis of *P–h* curves to construct a set of dimensionless functions characterizing the sharp indentation of the instrument, to correlate the data obtained from nanoindentation to the elastic–plastic energy, and also compared the results obtained by forward and reverse algorithms with experimental data, respectively. Lee et al. [22] established a series of dimensionless functions to reduce the sensitivity of data and experimental errors and to improve the accuracy of nanoindentation reverse analysis by using only *P–h* curves to obtain the parameters $C$ and $W_p/W_t$ for prediction in the order of $E$, $\sigma_r$, $H$, $n$ and $\sigma_y$. and the residual indentation depth and maximum indentation depth after unloading obtained from the AFM observation after indentation experiments were obtained by another method to obtain the hardening index $n$. Jiang et al. [36] described the elastoplastic properties of thin-film materials on elastoplastic substrates and proposed the following elegant equation as:

$$P = P_m \left( \frac{h}{h_m} \right)^x . \tag{1}$$

The maximum indentation load $P_m$ and the index $x$ are the functions of all independent parameters, $h$ is the indentation depth, and $h_m$ is the maximum depth of the indentation. Equation (1) was subjected to dimensionless analysis to obtain the dimensionless function and obtain the elastoplastic properties of the material. Yu et al. proposed a dimensionless equation in which the dimensionless function was established as a function of the indentation load $P$ and the contact depth $h_c$ as a function of the other parameters. Both dimensionless functions proposed by Jiang and Yu established two separate functions related to the other parameters to obtain the predicted *P–h* curves. Honga et al. [37] proposed a new analytical method based on the method of Dao et al. [35] by establishing a generalized dimensionless function and constructing a forward analysis algorithm and a reverse analysis algorithm, respectively, for directly predicting the nanoindentation response based on known elastoplastic properties. Based on FE simulations of nanoindentation, Long et al. [38] derived a dimensionless function and proposed a reverse algorithm for estimating the intrinsic parameters and surface stresses of elastoplastic materials. In addition, they used this reverse algorithm to estimate the residual stresses in solder samples under

different annealing conditions. Subsequently, Long et al. [38] combined the dimensionless method and FE simulations to consider the predicted nanoindentation response in the presence or absence of prestressing two cases to investigate the elastoplastic properties of the material and derived the dimensionless function of:

$$\frac{P}{\sigma_y h^2} = \Pi\left(\frac{E^*}{\sigma_y},\ n, \varepsilon_{pre}\right),\tag{2}$$

where $E^*$ and $\varepsilon_{pre}$ are the reduced modulus and pre-strain, respectively.

Ehsan et al. [39] suggested that an optimal approach to measuring the elastoplastic properties of materials is to combine experimental results, FE simulations, and dimensionless methods. To be more applicable in various complex cases, they proposed a method with fewer dimensionless functions to obtain the elastoplastic properties of materials. Ehsan Bazzaz et al. also proposed the minimum resultant error method to obtain the strain-hardening index by combining the errors of two dimensionless functions to extract the yield stress and strain-hardening index in the form of a unique measure. Based on previous studies, Wang et al. [40] proposed the dimensionless functions corresponding to different residual stresses of the material elastoplastic properties. They proved experimentally that the relationship between the residual stress and the dimensionless functions is linear. Long et al. [41] successfully obtained the complete constitutive relationship of different materials by investigating the reverse algorithm of nanoindentation to analyze the elastoplastic material properties by adopting a common assumption for isotropic materials, in which the stress–strain properties are assumed to satisfy a relationship of:

$$\sigma = \begin{cases} E\varepsilon & \varepsilon \leq \varepsilon_y \\ R\varepsilon^n & \varepsilon > \varepsilon_y \end{cases}.\tag{3}$$

As shown in Figure 4, the proposed reverse algorithm combines the elastic modulus of the material and the *P–h* curve obtained from the nanoindentation test to determine the constitutive relationship of the material. In a follow-up study, Long et al. [42] randomly selected the elastoplastic mechanical properties and film thickness. They used FE simulations to achieve a good agreement between the proposed reverse method for predicting the elastoplastic intrinsic curve and the stress–strain curve of the material. This provides theoretical guidance for an in-depth understanding of the exact relationship between the elastoplastic mechanical properties and the indentation response of thin-film materials.

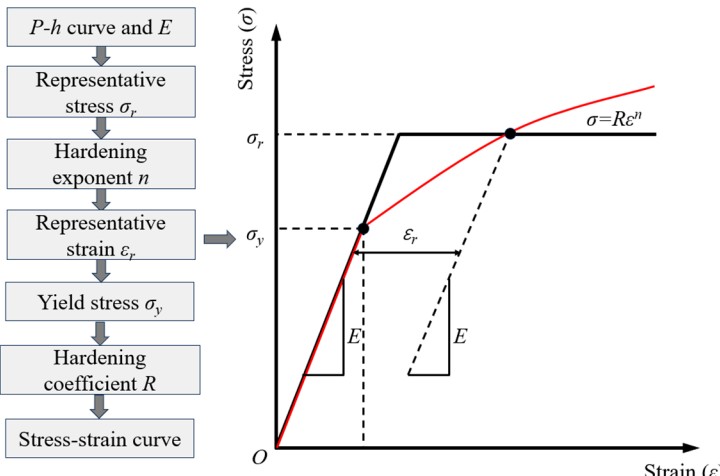

**Figure 4.** Schematic procedure of the proposed reverse analysis algorithm [41].

To obtain high-precision elastoplastic parameters for indentation depths below 100 nm, Sanchez-Camargo et al. [43] determined the elastoplastic parameters by reverse analysis,

and the numerical models all correctly described the indentation curves. For the measurement of elastoplastic properties of a ductile film on a hard substrate, Xing et al. [44] were able to provide sufficient information in a reverse analytical dimensionless algorithm at different indentation depths through a nanoindentation analysis method. In summary, combining the dimensionless method with nanoindentation is important for measuring the elastoplastic properties of materials and contributes significantly to material characterization.

### 2.2. Machine Learning

With the development of computer technology and artificial intelligence, combining machine learning and nanoindentation also inspires motivation to study material properties. Recently, Weng et al. [45] studied the material properties of cast iron based on machine learning and FE nanoindentation simulation and extracted the sharp stress–strain curve of cast iron by proposing the optimization algorithm particle swarm optimization (PSO) and the detailed steps of the stress–strain relationship inversion method are summarized in Figure 5.

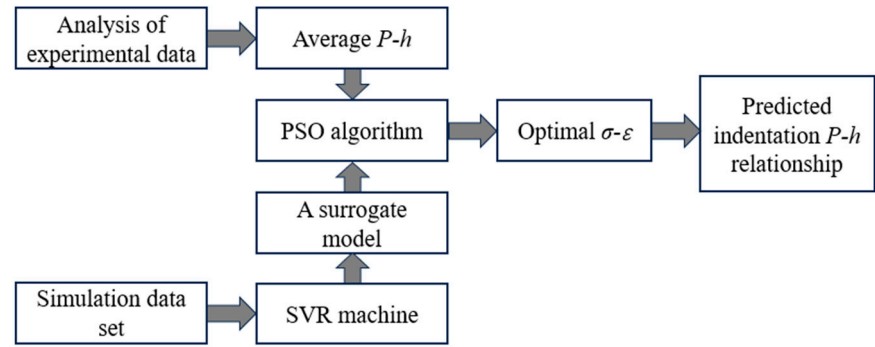

**Figure 5.** Steps of the inversion method of stress–strain relationship [45].

Laxmikant et al. [46] found that during the fabrication of electronic components, the mismatch in lattice and thermal expansion coefficients between the film and the substrate can lead to misfit strain. Therefore, finite element analysis (FEA)-based nanoindentation simulation methods and generative adversarial network (GAN)-based machine learning methods were used to predict the properties of thin-film layers. To study cracks generated by nanoindentation, Alipour et al. [47] combined a conceptual model of indentation with machine learning to determine fracture toughness using the energy method and *P–h* curves to fracture toughness to provide a systematic analysis for small-sized rock samples. Wang et al. [48] employed a hyperparametric tunable artificial neural network model to establish a positive relationship between the material elastoplastic parameters and the indentation *P–h* curve. Meanwhile, Long et al. [49] proposed a long short-term memory neural network to deeply learn the time series of *P–h* curves to predict the relationship between *P–h* curves of metal-coated materials and their stress–strain response. This network established the mapping relation from the *P–h* curves to the corresponding elastoplastic material stress–strain response. By analyzing the prediction results of the network, the final predicted *P–h* curves and stress–strain relationships match well with the power law equation. In addition, it is known that the material elastoplastic properties predicted by this method are more efficient and accurate than those obtained by FE analysis.

Further studies have demonstrated that complicated relationships exist between the indentation response and stress–strain curves of elastoplastic materials with thin-film substrates, posing challenges for conventional calculation methods. To overcome this problem, Long et al. [50] proposed a machine learning-based method, namely convolutional neural network (CNN), to rapidly obtain the mechanical properties of thin-film elastoplastic materials. Compared with the traditional reverse algorithm, CNN excels in reducing

computational complexity and computational time and has higher prediction accuracy for the intrinsic parameters of thin-film elastoplastic materials.

### 2.3. FEMU Method

The finite element model update (FEMU) method is a technique used to update the FE model of a structure by adjusting the FE model parameters according to the discrepancies. Comparison between the measured data with the FE simulation results leads to more accurately simulating and predicting the behavior of the actual structure [51–54].

The FEMU method typically consists of: (1) Collection of actual measurement data. The actual displacement, strain, acceleration, and other data from the structure are collected. (2) Construction of an initial FE model. An initial FE model is constructed as a reference model. (3) Comparison between the measured data and simulation results. The measured data is compared with the FE simulation results, and the differences are evaluated. (4) Adjustment of the FE model. The FE model is adjusted according to the comparison results by performing appropriate parameter updating techniques.

After iterating through the cycle of the above four steps, the FE model can be enhanced to simulate and predict material parameters more accurately, which can improve the reliability and accuracy of the model. Fauvel et al. [51] simulated nanoindentation tests by the FEMU method combined with a two-dimensional axisymmetric FE model to identify both the elastoplastic properties of 100 nm amorphous alumina films and the plastic properties of their silicon substrates by only relying on the *P–h* curves.

The FEMU method estimates one or more parameters by combining an FE model and an iterative optimization algorithm with the parameter $\theta_0$ as a starting point and by the difference between the force $P$ generated by the FE simulation and the force $P_{exp}$ obtained from the experimental data. Thus, based on the least squares method, the problem to be solved now becomes a minimization problem governed by:

$$\hat{\theta} = \operatorname{argmin} \omega[P(h;\theta), P^{exp}(h)], \tag{4}$$

where *h* is the displacement and $\omega$ is the cost function to minimize defined by

$$\omega(\theta) = \frac{1}{2T} \sum_{k=1}^{T} \left( \frac{P_k(\theta) - P_k^{exp}}{P_{max}^{exp}} \right)^2, \tag{5}$$

where *T* is the number of data points, $P_k$ and $P_k^{exp}$ are the simulated force and the experimentally measured force, respectively. $P_{max}^{exp}$ is the maximum experimental force.

To find the stability of the solution and improve the quality of the solution to quantify the experimentally measured information, the identifiability indicator (*I*-index) is introduced as:

$$I = \log_{10} \left( \frac{\lambda_{max}}{\lambda_{min}} \right), \tag{6}$$

where $\lambda_{max}$ and $\lambda_{min}$ are the extremal eigenvalue of the matrix close to the cost function minimum.

Parameter identification in the crystal plasticity model is more complicated than ordinary elastoplastic parameters. To solve this troublesome problem, Renner et al. [53] proposed the crystal plasticity finite element modeling update (CPFEMU) method for parameter identification. The proposed method is established to define the best-fit reverse problem and facilitate a more practical approach.

## 3. Contact Stiffness

The continuous stiffness measurement (CSM) test is an experimental method used to characterize the surface properties of a material. It differs from traditional indentation hardness testing methods in that the CSM test uses a continuous loading and unloading method in which force and displacement are measured simultaneously with the application

of strain. The CSM test provides a stiffness curve for a material by continuously measuring force and displacement data at strain rates. The stiffness curve can reflect the deformation characteristics of the material during the loading process, including the elastic phase, the plastic phase, and possibly other phases. Through analyzing the slope and shape of the stiffness curve, the hardness, elastic modulus, and other mechanical properties of the material can be inferred. The CSM technique is widely practiced in materials science and engineering, especially for studying surface properties of thin films, coatings, nanomaterials, etc. [55–57]. Moreover, it can provide a non-destructive and rapid method to obtain mechanical parameters of materials, which can provide valuable information for material design, quality control, and material failure analysis. Therefore, extensive works have also been accomplished on obtaining other properties of materials through their stiffness.

In 2005, both Zhao and Ogasawara investigated the elastoplastic properties of materials obtained by a single indentation test [58,59]. Among them, Ogasawara obtained from the indentation work during loading and the contact stiffness during unloading by fitting the contact stiffness $S$ after normalization from the plane strain modulus E and the maximum displacement of the indenter to obtain:

$$\Omega \equiv \frac{S}{2\delta_{max}\overline{E}} = A\xi^3 + B\xi^2 + C\xi + D,$$ (7)

where $A$, $B$, $C$, and $D$ are the coefficients of the fitted equation.

Ogasawara et al. [58] completed 60 different combinations of material properties ($n$ = 0.0, 0.1, 0.2, 0.3, 0.4, 0.5) by a wide range of FE analysis, covering basically all engineering materials, establishing a clear relationship between indentation parameters and material properties. By detailed FE analysis of the indentation process in various materials, Rodríguez [60] obtained the actual hardness from the following dimensionless equation:

$$\frac{H}{\sigma_{yc}} = f\left(\frac{\sigma_{yc}}{E^*}, \beta\right),$$ (8)

where $\sigma_{yc}$ stands for the uniaxial compressive yield stress, $\beta$ is the pressure sensitivity index, and $E^*$ can be obtained from:

$$E^* = \frac{S\sqrt{\pi}}{2\sqrt{A}},$$ (9)

where $S$ is the unloading stiffness and $A$ is the actual contact area of the indentation. With Equation (8), problems, such as the extraction of elastic modulus and compressive yield strength from the instrumented indentation, can be better solved.

Furthermore, Long et al. [61] performed CSM-based nanoindentation experiments on Sn-3.0Ag-0.5Cu (SAC305) solder by defining the experimentally obtained contact stiffness to represent the residual stress in the solder samples after the cooling and annealing processes and using the Oliver–Pharr model to fit the unloading response of the indentation. It was found that for SAC305 solder samples with different cooling processes, annealing treatments lasting 6 h effectively reduced the residual stresses. Additionally, they performed nanoindentation experiments on pressureless sintered silver nanoparticle samples at room temperature [62]. To accurately measure the strain rate sensitivity, their research team adopted a novel technique to realize multiple strain rate jumps based on CSM. Through considering different strain rates and indentation depths in the experiments, the nanoscale mechanical properties were obtained. Phani et al. [63] developed a comprehensive model for simulating constant indentation strain rate in CSM tests to gain insight into the parameters that affect the precision and accuracy of the measurements. Based on the predictions of the proposed model, they proposed a new test method without closed-loop feedback and found that the method could significantly improve the precision and accuracy of CSM-based indentation measurements. The schematic diagram of the load time variation

during CSM-based indentation tests on the indenter head is shown in Figure 6. Through experiments and analysis based on CSM-based nanoindentation, Long et al. [64] investigated the mechanical properties and intrinsic behavior of SiC particle-reinforced sintered AgNP and proposed an analytical method to simulate the indentation behavior. The results reveal the correlation between microstructure and macroscopic properties, which guides the design of AgNP morphologies and improves the mechanical properties of sintered silver nanoparticles in the electronic packaging industry.

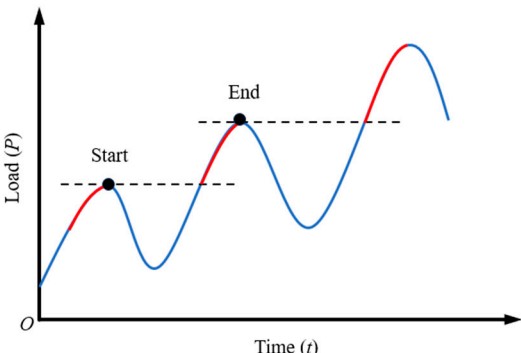

**Figure 6.** Schematic illustration of representative load–time variation on the indenter of CSM-based indentation test [63].

### 4. Density Functional Theory

Density functional theory (DFT) is an accurate, first-principle method for predicting material properties. In recent years, researchers have utilized DFT to calculate and predict the mechanical properties and anisotropy of different material systems.

Through the study of Shenoy et al. [65], we have seen that DFT can accurately predict Young's modulus of Li-Si alloys, which is an important indicator of the mechanical properties of the alloys. This provides a reliable example of using density functional theory to analyze and optimize the mechanical properties of other alloy materials. Lamuta et al. [66] used different exchange-correlation function approximations of DFT in their theoretical estimation of the combined indentation modulus of materials. This further demonstrates the flexibility and applicability of density functional theory in studying the mechanical properties of materials. In addition, the work of Zhou et al. [67] combines DFT with nanoindentation experiments to study the mechanical anisotropy of the energetic crystal FOX-7. This demonstrates the synergistic application of DFT and experimental studies for a more comprehensive understanding and explanation of the mechanical behavior of materials. Hayes et al. [68,69] introduced the multiscale orbital free density functional theory-localized quasicontinuum (OFDFT-LQC) approach to the problem of dislocation nucleation in metallic systems under the action of a real-size indenter. This problem is difficult to be covered by conventional density functional theory for such large-scale and complex systems due to the high computational cost. The innovation of the OFDFT-LQC method is to combine the first-principles OFDFT with the LQC evolution of the macroscopic system, which enables multiscale simulations. The introduction of the OFDFT-LQC method makes it possible to study the mechanical properties of more complex and practical material systems while improving the efficiency and accuracy of the calculations. With the continuous progress of technology and the improvement of the method, the density functional theory and its combination of nanoindentation experiments and multiscale simulations will achieve more breakthroughs and applications in the field of materials science.

### 5. Molecular Dynamics

Molecular dynamics is a computational method for modeling atomic and molecular motions using classical mechanical principles. In nanoindentation research, the molecular dynamics approach is important because it can simulate the mechanical behavior of materials at the atomic scale and provides new understanding and ideas for complex

indentation processes. Fang et al. [70] investigated the effect of temperature on the atomic scale nanoindentation process using a three-dimensional molecular dynamics model. They used a Morse potential function to simulate the interatomic forces between the sample and the tool and found that both Young's modulus and hardness become smaller as the temperature increases. These results make an important contribution to the understanding of the mechanical properties of materials in high-temperature environments. Li et al. [71] calculated the hardness of pure and alloyed gold on different crystalline surfaces using classical molecular dynamics simulations. Unlike the traditional force-displacement dependence, they analyzed the relationship between hardness and force, which provides a new idea to study the mechanical properties of materials. This demonstrates the versatility and flexibility of applying molecular dynamics methods in nanoindentation studies. Alexey et al. [72] carried out large-scale molecular dynamics simulations of nanoindentation of titanium crystals by considering different types of indenters, obtaining different load-displacement curves, and calculating the hardness and Young's modulus. This study demonstrated the dependence of the deformation of the crystal structure on the type of indenter. It provided important information to study the mechanical response of different materials in nanoindentation. In another study, Fang et al. [73] investigated the effects of indentation deformation, contact, and adhesion on multilayer films by molecular dynamics simulations. The results showed that the maximum load, plastic properties, and adhesion of the specimens increased with increasing indentation depth. This study provides important insights into the mechanical response of indented films. Chen et al. [74] investigated the nanoindentation process of amorphous alloys using molecular dynamics methods. They found that the radius of the indenter did not have a significant effect on the material properties, whereas a larger loading rate led to an increase in the hardness and elastic modulus of the material. In addition, the temperature has a significant effect on the material properties, as the load and hardness decrease with increasing temperature, while the elastic modulus tends to increase. This provides an important basis for considering the temperature factor in the study of mechanical properties of amorphous alloy materials.

As a result of these different research works, it can see that the molecular dynamics approach has a wide range of applications in nanoindentation studies. It can simulate the mechanical behavior of materials at the atomic level, providing important theoretical support for studying properties such as hardness, elastic modulus, and adhesion. Combined with other computational methods, such as density functional theory, we can gain a more comprehensive understanding of the mechanical properties of materials and provide more accurate and efficient tools for material design and application.

## 6. Other Methods

In recent years, many important explorations based on other methods have been conducted to obtain the elastoplastic properties of materials by nanoindentation measurements. X-ray and neutron diffraction are commonly used non-destructive testing techniques for determining microstresses in materials and are widely used in materials science and engineering to non-destructively analyze the crystal structure and microstress state of materials. By understanding the stress distribution of a material, the properties and behavior of the material can be better understood to optimize the design and improve the service life of the material [75,76]. The widespread use of X-ray and neutron diffraction methods makes the study of nanoindentation more accurate and in-depth. Additionally, the hardness map obtained from nanoindentation can also be used to analyze the mechanical properties of the material. The hardness map provides information about the spatial distribution of hardness on the material's surface, which can be used to infer the local mechanical properties of the material. By carefully analyzing the hardness map, it is possible to gain insight into the mechanical behavior of the material and provide important guidance and reference for the application and design of the material. It should be noted that the hardness map is a measurement of relative surface properties, and the understanding of the properties of the whole material needs to be combined with other testing methods and simulation

analysis [77,78]. In addition to X-rays, neutron diffraction, and hardness diagrams, there are many other ways to study the mechanical properties of materials.

Jacq et al. [79] proposed a method for obtaining local micro-yield stresses of materials by nanoindentation based on the relationship between the increasing maximum load and the residual displacement of the material under the continuous nanoindentation loading–unloading cycles. It is shown that the nanoindentation technique is not only applicable to the measurement of hardness and elastic modulus at the micron scale but can also reveal the relevant interfacial fracture mechanisms. Urena et al. [80] investigated the interfacial mechanical properties of short carbon fiber-reinforced AA6061 composites coated with different metal films using the nanoindentation technique. By performing nanoindentation experiments in different regions of the matrix/fiber interface, changes in the hardness and elastic modulus of the matrix could be accurately predicted. In addition, they accurately evaluated the fracture propensity of the interface and performed separate push-out tests for vertically aligned fibers with composite surfaces to measure the shear strength of the interface. Zhang et al. [81] proposed a method to distinguish plastic, elastic and viscoelastic deformations based on indentation tests to obtain the mechanical parameters of polymeric materials. As shown in Figure 7, In the fast loading/unloading steps (steps 1 and 2), elastoplastic deformation is dominant and causes negligible visco-elastic deformation only (relaxation in step 3). $P_{creep} < P_{max}$ is chosen to prevent further time–independent plastic deformation, ensuring the dominance of visco-elastic deformation in the reloading and creep steps.

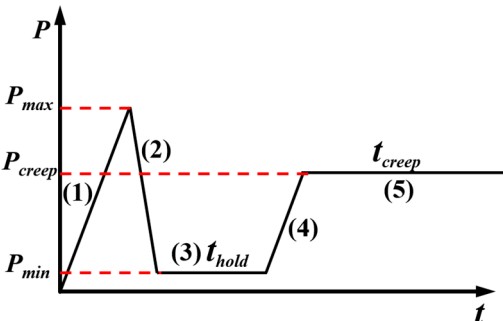

**Figure 7.** Schematic illustration of the five-step test scheme [81], where stage (1) represents the fast loading step to the maximum load $P_{max}$, stage (2) represents the fast unloading step to the very small load $P_{min}$, stage (3) is held under $P_{min}$ for a period time, stage (4) is the fast reloading step to the creep test load $P_{creep}$, and stage (5) represents the final holding step during the period of $t_{creep}$.

In addition, Zhang et al. [81] predicted the viscoelastic parameters using a genetic algorithm in combination with the analytical solution. The results show that the elastic viscoelastic parameters can be uniquely determined by combining the effective indenter proposed by Sakai [82] with the indenter proposed by Pharr and Bolshakov [83]. Furthermore, the values of the viscoelastic parameters extracted with the effective indenter proposed by Pharr and Bolshakov were found to be independent of the reloading level. Saraswati et al. [84] focused on the cyclic nanoindentation experiments and found that more material information could be extracted by this technique, especially material properties not available in conventional tests. Ma et al. [85] found that materials with the same modulus of elastoplastic $E$ and stress–strain parameters have similar indentation loading curves, regardless of the variation of the strain-hardening index $n$. Based on their results, the proposed method is proved to be more convenient and effective when obtaining the elastoplastic properties of the materials. First, according to Figure 8, the slope of the elastic segment is the same for the same studied material with changing the strain-hardening index, which means that the change in the strain-hardening index does not affect the elastic modulus of the studied material, so the yield strength and elastic modulus need to be determined by varying the assumed stress over a wide range without considering

strain-hardening until the calculated results and the experimental loading curve match. Then, the strain-hardening index is assumed to take different values within a certain reasonable range, such as 0 to 0.6. Finally, the actual strain-hardening index is obtained by continuously adjusting the calculated unloading curve to match the experimental curve with the assumed strain-hardening index value, as shown in Figure 9 [85].

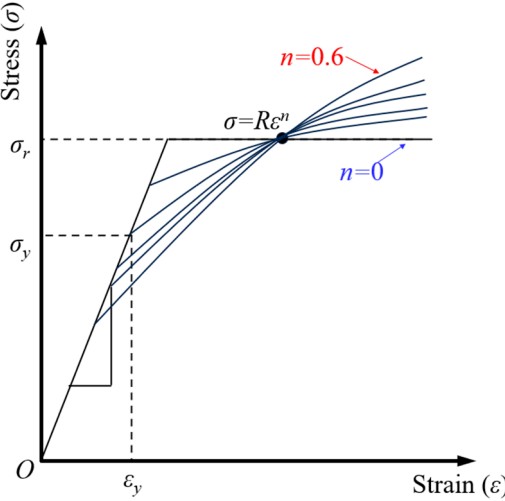

**Figure 8.** Schematic illustration of power-law stress–strain behavior of a material [85].

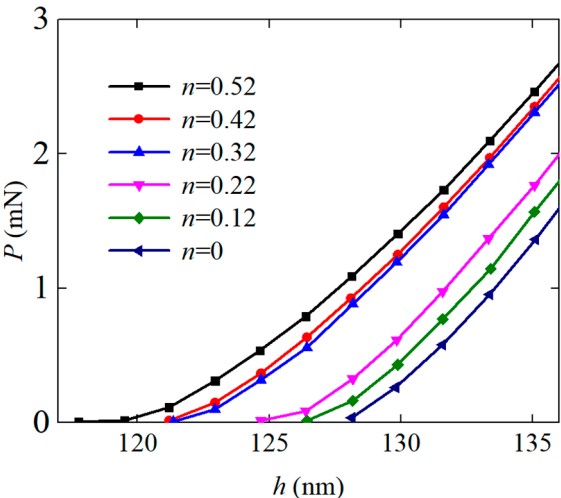

**Figure 9.** Partial comparison of the *P–h* curve with different assumed values of strain-hardening index *n* [85].

Weaver et al. [86] combined spherical nanoindentation and electron backscatter diffraction to characterize the elastic and plastic anisotropy of single crystals for two different compositions. Their results show that this proposed method can reliably characterize the elastic and plastic anisotropy of crystals with different alloy compositions. Roa et al. [87] discovered the linear relationship between the ratio of indentation hardness $H$ to the reduced modulus $E_r$ and the ratio of the elastic potential energy $U_e$ to the total strain energy $U_t$. This finding simplifies the data analysis using an independent analysis procedure, which resolves the considerable uncertainty due to the possible stacking effect of different plastic deformation phenomena.

It is important to acquire accurate material data with high precision and quality. However, during the experimental process of performing nanoindentation, the accumulation around the indenter leads to a significant increase in the contact area for large materials, while thin-film materials can lead to variations in the contact area due to deformation mech-

anisms as well as dislocations. In this regard, more advanced techniques, such as atomic force microscopy (AFM) or scanning electron microscopy (SEM), are needed to determine the contact area [85]. To obtain more accurate data, numerous researchers characterized the shape and size of the indentation based on AFM [85,88,89]. For the effect of the contact-area error caused by the indenter, the larger the radius, the greater the contact-area error of the indenter. To address this problem, Guo et al. [90] proposed a hardness determination method that is not based on the geometric relationship of the indenter tip. This method is highly effective regardless of the variation in the scale and depth size of the indentation. Lee et al. [91] also investigated the effects of material process hardening phenomena, such as indentation pile-up and sink effects, that are challenging to be explained by a simplified indentation resolution.

To improve the accuracy of the requested parameters depending on the way the material and indenter parameters are combined, Hyuk Lee proposed a more efficient reverse analysis method based on dimensional function analysis and artificial neural networks to build a database of dimensional functions related to material stress and strain parameters. The feasibility of the method was demonstrated by validating experimental results. In addition, the bimaterials consisting of matrix and particle make an important contribution to the development of new materials and the improvement of the properties of existing materials by jointly exploiting the interaction between matrix and particle. Therefore, it is of great importance in a wide range of applications. H.S. Tran proposed a new method for identifying the mechanical behavior of individual phases in a bimaterial material. An identification model was developed by performing nanoindentation tests at different locations of the nanocomposite using Berkovich indenters, which allowed the determination of material parameters and demonstrated the effect of particles on the nanoindentation response [92].

## 7. Current and Emerging Applications

To date, nanoindentation has become an effective method for studying the mechanical properties of materials such as metals, ceramics, polymers, and composites through the advantages of high resolution, non-destructive measurements, rapid measurements, comprehensive performance evaluation, and microscopic morphological analysis. It has been widely used in materials science, medicine, and manufacturing [93–95]. Stoichiometry is of paramount importance in the study of metals, metal alloys, and composite samples. Deviations from stoichiometric ratios or the presence of oxygen anions in the chemical composition of the samples will lead to changes in the cation charge state, which can significantly affect the electronic parameters [96,97]. Therefore, ensuring the accuracy of stoichiometry and avoiding the presence of anomalous oxygen anions is essential to safeguard the stability and superior performance of these materials to ensure optimal expression of their properties and performance. In addition, nanoindentation can position the indenter at specific locations on the material under test when measuring material properties, which helps to probe the nature of microstructural components and thus assess structural inhomogeneities [98,99].

Liu et al. [17] utilized the nanoindentation technique to verify its applicability in studying the nanomechanical properties of shale samples. The elastic modulus and hardness of different samples were calculated and compared using this method and correlated with mineral composition and microstructure. This study reveals the potential application of nanoindentation theory to shale samples and provides important information for understanding the mechanical behavior of shales. It can be predicted that the mechanical properties of shale samples can be studied in depth by nanoindentation methods, which can provide valuable references for shale oil and gas extraction and reservoir evaluation [17,100]. To study the degree of degradation of multilayered shales, the CSM method was used during the measurement of mechanical properties by applying small harmonic forces on the indenter and measuring the harmonic response of the indenter at the excitation frequency, which does not cause different types of damage to the rock surface and the rock

interior [55,101–103]. In addition, researchers have paid special attention to the study of the influence of micro- and macro-porous structures in gypsum materials on their mechanical properties and fracture mechanics through indentation experiments. This research allows insight into the fundamental mechanical behavior of highly complex materials used in various applications, such as the construction and medical industries [104].

In addition to conventional engineering sectors, nanoindentation has been proven to be one of the most promising methods for studying emerging areas such as bone tissue engineering. Nanoindentation is superior for providing information related to the nanomechanical properties of bone at the level of individual bone cells and bone lamellae, independent of their size, shape, and porosity [105–107]. This technique provides a more convenient method for medical research to study skeletal aging and the effects of various diseases on the bone. As such, it has potential applications in clinical research, such as understanding the impact of therapeutic processes. Further, microscopic information about the mechanical properties of bones can be obtained through nanoindentation techniques, providing insight into the structure and function of bones and offering new perspectives for the diagnosis and treatment of skeletal-related diseases [103].

In conclusion, nanoindentation has a wide range of applications in materials science and engineering and has promising prospects for the future. Combining nanoindentation with other characterization methods, such as atomic force microscopy (AFM) and scanning electron microscopy (SEM), thermal analysis methods, X-ray and neutron diffraction techniques, cell biology, and biocompatibility testing, and macro-mechanical testing methods can provide more comprehensive and multidimensional information about the properties of materials, thus deepening the understanding of materials. Multidimensional material characterization by nanoindentation, in combination with other characterization methods, is expected to address current challenges in materials research and play a greater role in materials design and applications.

## 8. Conclusions

Nanoindentation is an effective method for evaluating the elastoplastic properties of materials and is of great importance in materials science and engineering. However, the experimental data, such as the load-displacement curves and contact stiffness obtained from conventional measurements, are limited to analyzing the mechanical behavior of the indentation process. Therefore, novel numerical methods are critically reviewed in this paper for their advantages when combined with the nanoindentation to further reveal the mechanical behavior of materials. Among the representative numerical methods, the most common method is dimensionless analysis to reproduce the experimental results from different samples and under experimental conditions, which has the advantage of making the test results more generalizable. In addition, with the advantages of machine learning methods and finite element simulations, numerically predicted results can satisfactorily agree with actual indentation test data, thus helping to address the complexity of data analysis and interpretation in nanoindentation testing. Both machine learning methods and finite element model update methods could be good alternatives for providing accurate and efficient numerical tools in practice. It is well-accepted that nanoindentation has wide applicability in measuring a wide range of materials. It provides a reliable and accurate method for the mechanical property evaluation of different materials, including metallic materials, alloys, and composite materials. It is worth noting that in future developments regarding smaller scales and harsher loading environments, nanoindentation will face greater challenges and opportunities in terms of automation, versatility, and integration with other characterization techniques, as well as accurate mechanical models and algorithms. These developments will help further promote the application prospects and research areas of nanoindentation.

**Author Contributions:** Conceptualization, X.L. and R.D.; methodology, X.L. and R.D.; investigation, X.L. and R.D.; data curation, Y.S. and C.C.; writing—original draft preparation, X.L. and R.D.; writing—review and editing, Y.S. and C.C.; visualization, R.D.; supervision, X.L.; funding acquisition, X.L. All authors have read and agreed to the published version of the manuscript.

**Funding:** This work was funded by the National Natural Science Foundation of China (No. 52175148) and the Regional Collaboration Project of Shanxi Province (No. 202204041101044).

**Institutional Review Board Statement:** Not applicable.

**Informed Consent Statement:** Not applicable.

**Data Availability Statement:** Not applicable.

**Conflicts of Interest:** The authors declare no conflict of interest.

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
