# Peer review of "Critical Review of Nanoindentation-Based Numerical Methods for Evaluating Elastoplastic Material Properties"

_coatings, doi:10.3390/coatings13081334_

Round 1

Reviewer 1 Report

Dear Editor, 

Thanks. 

The review must be contained more survey related to the subject of study. 

Regards, 

Good

Author Response

Thank the reviewer for the valuable comment. The authors are very much in favor of including more survey-related content in the study. The authors have revised the paper accordingly to ensure that the literature review is more comprehensive and relevant to the topic of the study. Hope the reviewer will approve our improvements.

Reviewer 2 Report

The manuscript describes the current state of the art of the nanoindentation-based numerical methods, and therefore it may not be intended to show new results, which is acceptable for a review paper. Moreover, the content is well organized from the definition, mechanism, numerical study, and finally the application. However, the manuscript contains some confusion, as some are stated below.:

1.     The review paper failed to include some numerical methods such as molecular dynamics (MD), density functional theory (DFT) etc as the topic of this review paper suggested (Critical review of nanoindentation-based numerical methods).  In fact, many numerical results are available in the publications, and they are consistent with part of the results reported in this review. Why does the author choose to report on machine learning and FEMU only?

2.     The manuscript used some abbreviations of technical terms. Those abbreviations must be claimed in full form when it is used for the first time. For example, AFM, PMMA, FEMU, PSO, etc

3.     In section 1, line 71: “…stress-strain relationship of the material, are focused…”this sentence is hard to understand, it should be reconstructed.

4.     Figures 1, 3, 8, and 9 are not referenced. Are these authors’ own work?

5.     Can authors include some future prospects of nanoindentation of elastoplastic material properties into section 5: Current and emerging applications?

6.     A careful reading of the text should be done to suppress typo errors; can you check them, please? 

A careful reading of the text should be done to suppress typo errors; can you check them, please?

Author Response

A reply letter with detailed responses is attached as follows. Many thanks.

Reviewer 3 Report

Referee Report

On the paper “ Critical review of nanoindentation-based numerical methods for evaluating elastoplastic material properties “ (coatings-2511997) by the authors Xu Long, Ruipeng Dong, Yutai Su, and Chao Chang submitted to the Coatings

This is interesting revew paper. It reports the recent progress of nanoindentation-related researches and critically reviews the various numerical methods for evaluating elastoplastic constitutive properties of materials based on nanoindentation technology, which aims to provide a comprehensive understanding of the application and development trend of nanoindentation technique and to provide guidance and reference for further researches and applications. The reviewed experimental results are reliable without any doubts. However, I have some questions and additions. I would like to note a few points to improve the paper before it can be published:

1.    Everything the motivation should be deleted from the Abstract.

2.    The authors should give in 1. Introduction examples of the formation of thin films:

(1). A.L. Kozlovskiy, M.V. Zdorovets, Synthesis, structural, strength and corrosion properties of thin films of the type CuX (X = Bi, Mg, Ni), J. Mater. Sci.: Mater. Electron. 30 (2019) 11819-11832. https://doi.org/10.1007/s10854-019-01556-x.

(2). A. Kotelnikova, T. Zubar, T. Vershinina, M. Panasiuk, O. Kanafyev, V. Fedkin, I. Kubasov, A. Turutin, S. Trukhanov, D. Tishkevich, V. Fedosyuk, A. Trukhanov, Saccharin adsorption influence on the NiFe alloy films growth mechanisms during electrodeposition, RSC Adv. 12 (2022) 35722–35729. https://doi.org/10.1039/D2RA07118E.

3.    For metals, their alloys, and composite samples the stoichiometry is particularly important. The deviation from stoichiometry and appearance of the oxygen anions can lead to a change in the charge state of the cations, which in turn will greatly change the electronic parameters. That will seriously affect the practical application of the materials obtained. What is the oxygen stoichiometry of prepared samples? It is well known that the complex transition metal compounds easily allow the oxygen excess and/or deficit:

(3). S.V. Trukhanov, A.V. Trukhanov, A.N. Vasiliev, A.M. Balagurov, H. Szymczak, Magnetic state of the structural separated anion-deficient La0.70Sr0.30MnO2.85 manganite, J. Exp. Theor. Phys. 113 (2011) 819-825. https://doi.org/10.1134/S1063776111130127.

(4). A. Kozlovskiy, K. Egizbek, M.V. Zdorovets, M. Ibragimova, A. Shumskaya, A.A. Rogachev, Z.V. Ignatovich, K. Kadyrzhanov, Evaluation of the efficiency of detection and capture of manganese in aqueous solutions of FeCeOx nanocomposites doped with Nb2O5, Sensors 20 (2020) 4851. https://doi.org/10.3390/s20174851.

This should be discussed in 5. Current and emerging applications.

4.    The authors should mention in 1. Introduction and 4. Other methods such experimental techniques of non-destructive testing and determination of microstresses in materials as X-ray or/and neutron diffraction:

(5). A.V. Trukhanov, L.V. Panina, S.V. Trukhanov, V.G. Kostishyn, V.A. Turchenko, D.A. Vinnik, T.I. Zubar, E.S. Yakovenko, L.Yu. Macuy, E.L. Trukhanova, Critical influence of different diamagnetic ions on electromagnetic properties of BaFe12O19, Ceram Int. 44 (2018) 13520-13529. https://doi.org/10.1016/j.ceramint.2018.04.183.

(6). D.I. Shlimas, A.L. Kozlovskiy, M.V. Zdorovets, Study of the formation effect of the cubic phase of LiTiO2 on the structural, optical, and mechanical properties of LixTixO3 ceramics with different contents of the X component, J. Mater. Sci.: Mater. Electron. 32 (2021) 7410-7422. https://doi.org/10.1007/s10854-021-05454-z.

5.    The proposed 6 papers should be inserted in References.

The paper should be sent to me for the second analysis after the major revisions.

Minor editing of English language required

Author Response

(The authors gave the same response as above.)

Reviewer 4 Report

I would like to express my support for the publication of a research paper entitled "Critical review of nanoindentation-based numerical methods for evaluating elastoplastic material properties".

Nanoindentation is a well-established method for evaluating the elastic-plastic properties of materials and is of great importance in materials science and engineering. The article provides an overview of new numerical methods in combination with nanoindentation. The use of machine learning and modeling methods allows us to determine data that can satisfactorily match the actual data of indentation tests, which allows us to solve the problem of the complexity of data analysis and interpretation when testing for nanoindentation. Machine learning methods and finite element model updates can be good alternatives to provide accurate and efficient numerical tools in practice.

Despite the obvious advantages of the work, some important works are not described:

1.                 The text does not mention that not only the Berkovich form identifiers are used for indentation, but also others. For example, cube corner, Vickers, and Knoop nanoindenter, which are generally recognized standards due to their wide application and well-known properties. Similar geometries are in the works [Mohan, S., Millan-Espitia, N., Yao, M. et al. Critical Evaluation of Spherical Indentation Stress-Strain Protocols for the Estimation of the Yield Strengths of Steels. Exp Mech 61, 641–652 (2021). https://doi.org/10.1007/s11340-021-00689-7] and [Xingshuo Huang, Alan Salek, Andrew G. Tomkins, Colin M. MacRae, Nicholas C. Wilson, Dougal G. McCulloch, Jodie E. Bradby; Hardness of nano- and microcrystalline lonsdaleite. Appl. Phys. Lett. 20 February 2023; 122 (8): 081902. https://doi.org/10.1063/5.0138911].

2.                 Hardness maps is not described in the work. For example, published works [Eli Saùl Puchi-Cabrera, Edoardo Rossi, Giuseppe Sansonetti, Marco Sebastiani, Edoardo Bemporad, Machine learning aided nanoindentation: A review of the current state and future perspectives, Current Opinion in Solid State and Materials Science, Volume 27, Issue 4, 2023, 101091, https://doi.org/10.1016/j.cossms.2023.101091.] and [Abhijeet Dhal, Ravi Sankar Haridas, Priyanka Agrawal, Sanya Gupta, Rajiv S. Mishra, Mapping hierarchical and heterogeneous micromechanics of a transformative high entropy alloy by nanoindentation and machine learning augmented clustering, Materials & Design, Volume 230, 2023, 111957, https://doi.org/10.1016/j.matdes.2023.111957.]

Author Response

(The authors gave the same response as above.)

Round 2

Reviewer 2 Report

The authors have made all the necessary modifications in the revised manuscript. It can now be accepted in the current form

Necessary modifications made.

Reviewer 3 Report

Referee Report

On the paper “ Critical review of nanoindentation-based numerical methods for evaluating elastoplastic material properties “ (coatings-2511997-v2) by the authors Xu Long, Ruipeng Dong, Yutai Su, and Chao Chang submitted to the Coatings

This paper has been well corrected and it can be recommended.

Minor editing of English language required